# Association between diet-related greenhouse gas emissions and nutrient intake adequacy among Japanese adults

Minami Sugimoto[1], Kentaro Murakami[2], Aya Fujiwara[2,3], Keiko Asakura[4], Shizuko Masayasu[5], Satoshi Sasaki[1,2]*

1 Department of Social and Preventive Epidemiology, Division of Health Sciences and Nursing, Graduate School of Medicine, University of Tokyo, Tokyo, Japan, 2 Department of Social and Preventive Epidemiology, School of Public Health, University of Tokyo, Tokyo, Japan, 3 Department of Nutritional Epidemiology and Shokuiku, National Institute of Biomedical Innovation, Health and Nutrition, Tokyo, Japan, 4 Department of Environmental and Occupational Health, School of Medicine, Toho University, Tokyo, Japan, 5 Ikurien-Naka, Ibaraki, Japan

* stssasak@m.u-tokyo.ac.jp, sasakicrs@m.u-tokyo.ac.jp

**Data Availability Statement:** Data cannot be made publicly available as the database contains sensitive and identifying information. Restrictions were imposed by the Ethics Committee of the

## Abstract

### Objectives

A growing number of Western studies have been exploring sustainable and healthy dietary patterns that target to reduce diet-related greenhouse gas emissions (GHGE) and to achieve nutritional needs. However, research is limited among Asian populations, where food sources for diet-related GHGE differ from those in Western populations. This study aimed to investigate associations between diet-related GHGE and the prevalence of inadequate nutritional intake.

### Methods

A cross-sectional study was carried out among 392 healthy Japanese volunteers aged 20–69 years. Dietary intake was assessed by four-non-consecutive day diet record. Diet-related GHGE was estimated using the Global Link Input-Output model and adjusted for energy intake by residual method. Prevalence of inadequacy was defined as a percentage of participants with nutrient intake outside the Tentative Dietary Goal for Preventing Lifestyle-Related Disease or below the Estimated Average Requirement defined by the Dietary Reference Intakes (DRIs) for Japanese. The association between diet-related GHGE and the prevalence of inadequacy of the usual intake of each nutrient was examined using logistic regression models.

### Results

Participants with higher diet-related GHGE had overall better adherence to the DRIs. Intakes of all selected nutrients were positively associated with diet-related GHGE, except for carbohydrate, total fat, and saturated fat. With increasing quartile of diet-related GHGE, the prevalence of inadequacy decreased for protein, dietary fiber, potassium, vitamins A, B-

University of Tokyo, Faculty of Medicine. The non-author point of contacts for data access are as follows: 1) the Department of Social and Preventive Epidemiology, Division of Health Sciences and Nursing, Graduate School of Medicine, University of Tokyo (email: nutrepibox@m.u-tokyo.ac.jp) and 2) the Ethics Committee of the University of Tokyo, Faculty of Medicine (email: ethics@m.u-tokyo.ac.jp). Interested researchers may also contact the corresponding author Satoshi Sasaki (stssasak@m.u-tokyo.ac.jp).

**Funding:** The present study was supported by a Health and Labour Sciences Research Grant (H23-Jyunkankitou (seishuu)-ippan-001) from the Ministry of Health, Labour and Welfare, Japan and a Grant-in-Aid for Japan Society for the Promotion of Science Fellows (18J21618) from the Japan Society for the Promotion of Science.

**Competing interests:** The authors have declared that no competing interests exist.

6, and C, thiamin, riboflavin, calcium, magnesium, iron, and zinc, while that for sodium increased.

## Conclusions

Diets with lower diet-related GHGE did not have better adherence to the DRIs compared to diets with higher diet-related GHGE among Japanese adults. Drastic dietary change or other strategies such as improving the food system would be needed to achieve a sustainable and healthy diet among Japanese.

## Introduction

In the context of growing interest for sustainable healthy diets [1] and climate change, many epidemiological studies have been focused on individual dietary choice and greenhouse gas emissions (GHGE) related to the diet [2–21]. Individual dietary choices would consequently affect emissions from the food sectors because it drives food systems, which contribute 21–37% global GHGE when pre- and post-production stages are included [22]. Thus, it is expected that achieving sustainable healthy dietary choices at individual level would reduce GHGE from food sector.

Most of previous observational studies have shown inverse associations between diet-related GHGE and adequacy of nutrient intake [3] and overall diet quality [4–9], while a few showed positive or null associations [2, 7]. On the other hand, a series of scenario studies have shown that modeled healthy diets meeting dietary guidelines did not always improve diet-related GHGE [10–12]. However, these previous studies mainly come from Western countries [2–18], while research is limited among Asian countries including Japan, where meat intake is lower than Western countries [19–21].

Japan was the sixth-largest greenhouse gas emitter in 2016 [23]. The Japanese government advocated the strategies "to reduce greenhouse gas emissions 80% by 2050 as its long-term goal [24]." The environmental dimension of diet has not been included in this statement nor mentioned in the dietary guidelines. It might be the result of a lack of evidence about the environmental impact of Japanese diet, including diet-related GHGE. Japanese dietary habits have long been of interest to researchers from other countries from a nutrition and health standpoint [25, 26], while its environmental aspects have been rarely investigated.

The contemporary Japanese diet is typically high in refined grains, seaweeds, vegetables, fish, legumes and low in whole grains, nuts and seeds, dairy products, sugar-sweetened beverage, and processed and unprocessed red meats [27–29]. At the nutrient level, it is characterized by a high intake of sodium and a low intake of dietary fiber, calcium, and saturated fat [29, 30]. Major dietary sources of GHGE among Japanese included meat (19.7%), fish and seafood (13.8%), and cereals (13.1%) [19]. They are quite different from those observed in Western countries, where a considerably large proportion of diet-related GHGE is explained by meat (31.3–56.6%) and dairy products (13.1–25.2%), with a small contribution of fish and seafood (2.1–9.2%) and cereals (2.1–10.5%) [6, 13–18]. Adverse health effects of higher meat intake have been reported in a population with high meat intake [31], while negative or no associations between meat intake and mortality from cardiovascular disease or cancer were generally found in Asian population with low meat intake [31, 32]. Hence, the association between diet-related GHGE and dietary intake among Japanese could differ from that observed in Western

studies. Therefore, this study aimed to investigate associations between diet-related GHGE and the prevalence of inadequate nutritional intake in Japanese adults.

## Materials and methods

### Study design and participants

This cross-sectional study was based on data from healthy Japanese adults aged 20–69 years. Data collection was conducted in 20 study areas covering 23 of 47 prefectures between February and March 2013. Details of the study have been reported elsewhere [33, 34]. The primary objective of this survey was to estimate sodium and potassium excretion using biomarker and to identify food sources of sodium and potassium. First, 199 dietitians working in separate welfare facilities were recruited as research dietitians supporting the survey. Next, the research dietitians recruited participants from their co-workers or family members of co-workers with stratifying by sex and by five 10-year age bands (20–29, 30–39, 40–49, 50–59, and 60–69 years). The number of participants was targeted to be 40 adults in each study area to allow for statistical analysis stratified by sex, age, body mass index (BMI; in kg/m$^2$), and physical activity. The exclusion criteria were: (i) licensed dietary or medical provider, (ii) residence in the prefecture or adjacent prefecture in which the facility was located for less than 6 months, (iii) individuals who were under diet therapy prescribed by a doctor or dietitian at the time of the study or within 1 y before the study, (iv) pregnant or lactating women, and (v) individuals who had history of hospitalization for diabetes education. Of those 800 adults recruited, nine adults withdrew from the survey. In total, 791 adults participated. To reduce the burden to the participants and the research dietitians, half of the participants (n = 400) were also asked to complete diet records. A total of 392 adults (196 men and 196 women) completed diet records and were included in the present analysis. BMI was calculated based on measured weight and height. The participants' occupation, educational background, and smoking habits were assessed using a questionnaire (S1 Appendix in S1 File).

This study was conducted according to the guidelines laid down in the Declaration of Helsinki and all procedures involving research study participants were approved by the Ethics Committee of the University of Tokyo, Faculty of Medicine (approval number: 10005, approval date: January 7, 2013). Written informed consent was obtained from all participants.

### Dietary assessment

Dietary intake was assessed by four-non-consecutive-day diet records. The assessment days consisted of three working days and one day off. All participants were provided with digital kitchen scale (KD-812WH; Tanita, Tokyo, Japan), measuring spoon, measuring cup, a manual for the diet record, and recording sheets and instructed how to weigh and record foods and beverages consumed. Each participant was asked to weigh and record all food and beverages consumed on the four assessment days using the provided equipments and recording sheet. When weighing was difficult (e.g. eating out), the restaurant's name, name of dishes, and an estimated amount of leftovers were reported. Pictures of food and beverages were also provided by some participants but not mandatory. All recorded foods and beverages were assigned food item numbers according to the Standard Tables of Food Composition in Japan, Fifth Revised and Enlarged Edition [35]. All records were checked twice (by the research dietitians at each facility and trained dietitian staff at the survey center). The research dietitian contacted the participants to clarify any ambiguities or missing data in the recording sheets. Daily intakes of foods, energy, and nutrients were estimated based on the Standard Tables of Food Composition in Japan in which free sugar content of each food item was added based on

published sources [36, 37] due to the insufficiency of data for free sugars in original food composition tables.

To evaluate the accuracy of the reported energy intake (EI), the ratio of EI to basal metabolic rate (BMR) (EI:BMR) was compared to the Goldberg cut-off [38]. For this, the average EI of the four-assessment-days was used. BMR was calculated using the sex-specific equation for Japanese population based on age, weight, and height. Assuming sedentary lifestyle for all subjects because of a lack of objective information on physical activity in this study (physical activity level: 1.55 for men and 1.56 for women), under-, plausible-, and over-reporters were defined as having EI:BMR <1.02, 1.02–2.35, and >2.35 for men, and <1.03, 1.03–2.36, and >2.36 for women, respectively. Under- and over- reporters were identified in this study but were not excluded from the analysis to avoid bias for that exclusion. Assuming sedentary lifestyle for all subjects because of a lack of objective information on physical activity in this study (physical activity level: 1.55 for men and 1.56 for women), under-, plausible-, and over-reporters were defined as having EI:BMR <1.02, 1.02–2.35, and >2.35 for men, and <1.03, 1.03–2.36, and >2.36 for women, respectively [39, 40].

## Greenhouse gas emissions of food and drinks

GHGE of foods and drinks were estimated using the food GHGE database based on the Global Link Input-Output (GLIO) model [41]. The detailed description of the database development was written elsewhere [19]. Briefly, the GLIO model includes 804 economic sectors in Japan and 230 foreign countries and regions describing the relationship between their production and consumption systems. Production-based GHGE values for 354 food items were calculated from the GLIO model. Then, the values of 354 foods and drinks were systematically linked to food items from the Standard Tables of Food Composition in Japan [35]. Diet-related GHGE was calculated by multiplying the GHGE value for food items and the mean food intake of the four assessment days.

## Estimation of usual nutrient and food intakes and greenhouse gas emissions

The usual dietary intake of nutrients, foods, and diet-related GHGE were estimated with the Multiple Source Method (MSM) [42, 43]. For each participant, four-day measurements of food intake, nutrient intake, or diet-related GHGE data were imported to the MSM program. In the MSM, usual intake for each participant was calculated as following three steps: calculating the probability of eating a certain food or nutrient on a random day was estimated for each individual, estimating the usual amount of food or nutrient intake on a consumption day, and multiplying resulting numbers from former two steps by each other. The usual percent of energy from protein, fat, carbohydrate, saturated fat, and free sugar was estimated by the MSM after calculation of percent energy from these nutrients for each assessment day.

## Assessment of nutrient intake inadequacy

For the nutrient intake inadequacy assessment, the estimated usual nutrient intakes were compared with age- and sex-specific reference values in the Dietary Reference Intakes (DRIs) for Japanese [44], as described in previous studies [45, 46]. In brief, several types of reference values are set according to their purposes in the Japanese DRIs. The Estimated Average Requirement (EAR) is defined as "the estimated intake amount that meets the requirements of 50% of the individuals belonging to an age or sex group [47]." The Tentative Dietary Goal for Preventing Lifestyle-Related Disease (DG) is defined as "the nutrient intake amount Japanese people should aim for, to prevent lifestyle-related diseases, in the foreseeable future [47]." The DGs

are provided for macronutrient balance (% of energy from protein, total fat, saturated fat, and carbohydrates), dietary fiber, sodium, and potassium. For seven nutrients with DGs, participants whose intake levels were outside the range of the corresponding DG values were considered inadequate. For nutrients with EAR except for iron, participants whose intake was lower than the EAR were considered inadequate. Because of the strongly skewed distribution of iron requirements among menstruating women [48, 49] and the lack of information about menopause among female participants, less than 9.3 mg/d of iron intake (probability of inadequacy >50% for menstruating women whose bioavailability of iron is 15%) [50] was considered as inadequate among women under 50 years old using a probability method. For free sugar, the participants whose intake was 5% energy or more were deemed to be inadequate based on the conditional recommendation advocated by the World Health Organization (WHO) [51] because there is no reference value for free sugar in Japanese DRIs. The reference values are shown in S1 Table in S1 File.

We excluded biotin, iodine, selenium, chromium, and molybdenum for which EAR is set from the present analysis because of insufficiency of food composition tables for these nutrients in Japan.

## Statistical analysis

All statistical analyses were performed with the SAS statistical software, version 9.4 (SAS Institute Inc., Cary, NC, USA). All reported P-values were two-tailed, with a P-value < 0.05 considered statistically significant. The mean and standard deviation (SD) of the usual nutrient intake and prevalence of inadequacy among all participants were described. Because energy intake was highly correlated with both nutrient intakes and diet-related GHGE, usual diet-related GHGE was adjusted for energy by residual method. Energy-adjusted diet-related GHGE was computed as the residuals from the regression model, with total energy intake as the independent variable and absolute diet-related GHGE as the dependent variable [52]. The participants were divided into sex-specific quartiles (i.e. divided separately by sex) according to the usual diet-related GHGE (g $CO_2$-eq/d), and then mixed for further analysis as our sample size was small, and the analysis by sex showed similar results (S2–S6 Tables in S1 File).

Linear regression models were constructed to examine the association between diet-related GHGE (g $CO_2$-eq/d) and age and BMI using the median value for each quartile as a continuous variable. Mean differences of age and BMI among quartile groups were also tested by one-way analysis of variance (ANOVA) with the *post hoc* Bonferroni test. The chi-square test was used to test differences in living area, occupation, educational background, and smoking habits.

Linear regression models were constructed to examine the association between diet-related GHGE (g $CO_2$-eq/d) and the usual nutrition intake using the median value for each quartile as a continuous variable. Differences between quartiles were also assessed using one-way ANOVA with post hoc Bonferroni's test. Next, using the logistic regression, the odds ratio (OR) and 95% confidence interval (CI) for inadequacy were calculated for each quartile of diet-related GHGE, with the lowest quartile category used as the reference. Further, linear trends of OR were tested with increasing levels of diet-related GHGE by assigning each subject the median value for the category and modeling this value as a continuous variable. For the nutrients of which the prevalence of inadequacy among all participants was less than 10%, the odds ratio and linear trends of OR were not analyzed due to a small prevalence of each quartile group. In addition, the association between diet-related GHGE and the usual food intake was examined using linear regression models and one-way ANOVA with post hoc Bonferroni's test.

**Table 1. Basic characteristics of participants according to quartile (Q) of diet-related GHGE (g CO$_2$-eq/day) among 392 Japanese adults[a].**

| | All (n = 392) | Q1 (n = 98) | Q2 (n = 98) | Q3 (n = 98) | Q4 (n = 98) | P[c] |
|---|---|---|---|---|---|---|
| | | 3241 (2992, 3550)[b] | 3757 (3450, 4044)[b] | 4191 (3844, 4511)[b] | 4833 (4420, 5198)[b] | |
| Age (years) | 44.5 ± 13.4 | 39.6 ± 13.1* | 44.9 ± 13.6†‡ | 44.1 ± 13.1*† | 49.4 ± 11.9‡ | <.0001 |
| Body mass index (kg/m$^2$) | 23.3 ± 3.6 | 23.0 ± 3.8 | 23.6 ± 4.0 | 22.9 ± 3.1 | 23.7 ± 3.6 | 0.39 |
| Living area (%) | | | | | | 0.02 |
| Hokkaido and Tohoku | 15.1 | 21.4 | 18.4 | 7.1 | 13.3 | |
| Kanto | 20.2 | 20.4 | 24.5 | 22.4 | 13.3 | |
| Hokuriku and Tokai | 9.4 | 10.2 | 10.2 | 12.2 | 5.1 | |
| Kinki | 15.1 | 13.3 | 7.1 | 16.3 | 23.5 | |
| Chugoku and Shikoku | 20.2 | 18.4 | 23.5 | 21.4 | 17.3 | |
| Kyusyu and Okinawa | 20.2 | 16.3 | 16.3 | 20.4 | 27.6 | |
| Occupation (%) | | | | | | 0.003 |
| Clerical | 41.8 | 35.7 | 40.8 | 49.0 | 41.8 | |
| Nursing care | 41.8 | 52.0 | 41.8 | 33.7 | 39.8 | |
| Medical assistant | 3.1 | 2.0 | 0.0 | 9.2 | 1.0 | |
| Cooking assistant | 6.1 | 4.1 | 7.1 | 3.1 | 10.2 | |
| Others | 7.1 | 6.1 | 10.2 | 5.1 | 7.1 | |
| Educational background (%) | | | | | | 0.82 |
| Junior high school or other | 2.6 | 5.1 | 1.0 | 2.0 | 2.0 | |
| Senior high school | 26.5 | 26.5 | 27.6 | 26.5 | 25.5 | |
| Vocational school or junior college | 36.7 | 38.8 | 37.8 | 33.7 | 36.7 | |
| University or graduate school | 34.2 | 29.6 | 33.7 | 37.8 | 35.7 | |
| Smoking habit (%) | | | | | | 0.70 |
| Nonsmoker | 56.1 | 54.1 | 58.2 | 56.1 | 56.1 | |
| Past smoker | 18.1 | 15.3 | 15.3 | 19.4 | 22.4 | |
| Current smoker | 25.8 | 30.6 | 26.5 | 24.5 | 21.4 | |

GHGE, greenhouse gas emissions; CO$_2$-eq, carbon dioxide equivalents.

*†‡ Maen values within a row with different symbols were significantly different between the quartile group by *post hoc* Bonferroni's test (P<0.05).

[a] Participants (196 men and 196 women) were divided into quartiles by usual diet-related GHGE separately by sex, and then combined for analysis. Usual diet-related GHGE was calculated using the Multiple Source Method [42, 43] and then adjusted for energy intake by residual method. Values are means ± SDs unless otherwise indicated.

[b] Usual diet-related GHGE (g CO$_2$-eq/d): median (25th, 75th percentiles).

[c] Trend of association was examined for age and body mass index using a linear regression model with the median value of diet-related GHGE in each quartile as a continuous variable. χ2 test was used for categorical variables.

## Results

The basic characteristics of the participants are described in Table 1. The mean age was 44.5 years and mean BMI was and 23.3 kg/m$^2$. Around 80% of the participants worked on clerical or nursing care. The prevalence of under-reporters (defined as having EI:BMR <1.02 and for men and <1.03 for women) was 3.6% (nine men and five women) and that of over-reporters (defined as having EI:BMR >2.35 for men and >2.36 for women) was 2.3% (three men and six women). Participants with higher diet-related GHGE were significantly older than those with lower diet-related GHGE.

The overall prevalence of inadequacy is shown in Table 2. Of seven nutrients with DG, over 60% of the participants did not meet the DRIs for saturated fat, dietary fiber, sodium, and potassium. In particular, 90% of the participants did not meet the DRIs for dietary fiber and sodium. Similarly, 72% of participants did not meet the WHO's conditional recommendation

**Table 2. Usual nutrient intake according to quartile (Q) of diet-related GHGE (g CO$_2$-eq/day) among 392 Japanese adults (aged 20–69 y)[a].**

| | All (n = 392) | | Q1 (n = 98) | Q2 (n = 98) | Q3 (n = 98) | Q4 (n = 98) | P for trend[c] |
|---|---|---|---|---|---|---|---|
| | | Participants outside or below reference value (%) | 3241 (2992, 3550)[b] | 3757 (3450, 4044)[b] | 4191 (3844, 4511)[b] | 4833 (4420, 5198)[b] | |
| Energy (kcal/d) | 2115 ± 434 | - | 2111 ± 467 | 2109 ± 438 | 2092 ± 441 | 2149 ± 390 | 0.58 |
| Nutrients with DG | | <DG >DG | | | | | |
| Protein (% energy) | 14.1 ± 1.4 | 31.9 - | 13.2 ± 1.3* | 14.0 ± 1.2*† | 14.4 ± 1.4†‡ | 14.9 ± 1.2‡ | <.0001 |
| Total fat (% energy) | 27.9 ± 3.8 | 2.3 28.1 | 27.5 ± 3.9 | 28.6 ± 4.2 | 27.5 ± 3.5 | 27.9 ± 3.5 | 0.13 |
| Saturated fat (% energy) | 8.1 ± 1.6 | - 75.3 | 7.9 ± 1.8 | 8.3 ± 1.7 | 8.0 ± 1.6 | 8.1 ± 1.3 | 0.28 |
| Carbohydrate (% energy) | 53.6 ± 5.4 | 22.7 1.0 | 55.2 ± 5.0* | 53.5 ± 5.4*† | 53.0 ± 5.7*† | 52.5 ± 4.9† | 0.003 |
| Dietary fiber (g/d) | 13.9 ± 4.1 | 91.8 - | 12.4 ± 3.7* | 13.8 ± 4.1* | 13.9 ± 3.2*† | 15.6 ± 4.5† | <.0001 |
| Sodium (g NaCl equivalent/d) | 10.2 ± 2.4 | - 92.6 | 9.7 ± 2.3* | 10.0 ± 2.3*† | 10.0 ± 2.1*† | 10.9 ± 2.8† | 0.002 |
| Potassium (mg/d) | 2638 ± 663 | 62.5 - | 2292 ± 575* | 2578 ± 583† | 2659 ± 585† | 3021 ± 698‡ | <.0001 |
| Nutrient with the World Health Organization's conditional recommendation | | ≥5% Energy | | | | | |
| Free sugar (% energy) | 6.9 ± 3.0 | 71.6 | 7.3 ± 3.9 | 6.4 ± 2.3 | 7.4 ± 3.0 | 6.6 ± 2.5 | 0.04 |
| Nutrients with EAR | | <EAR (%) | | | | | |
| Protein (g/d) | 74.0 ± 15.7 | 1.5 | 68.2 ± 14.5* | 73.4 ± 15.7*† | 74.5 ± 15.6*† | 79.8 ± 15.0† | <.0001 |
| Vitamin A (µg RAE/d) | 524 ± 213 | 61.5 | 431 ± 146* | 525 ± 230† | 542 ± 160† | 596 ± 262† | <.0001 |
| Thiamin (mg/d) | 1.0 ± 0.2 | 60.2 | 0.9 ± 0.2* | 1.0 ± 0.2*† | 1.0 ± 0.2*† | 1.0 ± 0.2† | 0.0008 |
| Riboflavin (mg/d) | 1.3 ± 0.3 | 29.6 | 1.2 ± 0.3* | 1.3 ± 0.3*† | 1.3 ± 0.3*† | 1.4 ± 0.3† | <.0001 |
| Niacin (mg/d) | 18.7 ± 4.8 | 4.3 | 16.3 ± 4.4* | 17.8 ± 3.8*† | 19.4 ± 4.9†‡ | 21.2 ± 4.7‡ | <.0001 |
| Vitamin B-6 (mg/d) | 1.3 ± 0.3 | 24.5 | 1.1 ± 0.3* | 1.2 ± 0.3*† | 1.3 ± 0.3† | 1.5 ± 0.3‡ | 0.0008 |
| Vitamin B-12 (µg/d) | 6.3 ± 2.6 | 0.3 | 5.1 ± 2.0* | 6.3 ± 2.4† | 6.5 ± 2.6† | 7.2 ± 2.8† | <.0001 |
| Folate (µg/d) | 367 ± 124 | 6.1 | 304 ± 106* | 357 ± 120† | 374 ± 115† | 433 ± 123‡ | <.0001 |
| Vitamin C (mg/d) | 111 ± 45 | 31.1 | 90 ± 37* | 108 ± 38*† | 113 ± 41† | 135 ± 50‡ | <.0001 |
| Calcium (mg/d) | 509 ± 155 | 68.4 | 467 ± 158* | 521 ± 155*† | 502 ± 137*† | 547 ± 162† | 0.003 |
| Magnesium (mg/d) | 287 ± 75 | 41.1 | 258 ± 77* | 281 ± 68* | 288 ± 71*† | 319 ± 74† | <.0001 |
| Iron (mg/d)[d] | 8.3 ± 2.0 | 33.7 | 7.4 ± 2.0* | 8.1 ± 1.9*† | 8.3 ± 1.8†‡ | 9.2 ± 2.0‡ | <.0001 |
| Zinc (mg/d) | 8.6 ± 2.0 | 37.2 | 8.0 ± 1.9* | 8.6 ± 2.3*† | 8.5 ± 1.9*† | 9.3 ± 1.8† | <.0001 |
| Copper (mg/d) | 1.2 ± 0.3 | 1.8 | 1.1 ± 0.3* | 1.2 ± 0.3*† | 1.2 ± 0.3*† | 1.3 ± 0.3† | 0.003 |

GHGE, greenhouse gas emission; CO$_2$-eq, carbon dioxide equivalents; DG, Tentative Dietary Goal for Preventing Lifestyle-related Diseases; EAR, Estimated Average Requirement; RAE, retinol activity equivalent.

*†‡Maen values within a row with different symbols were significantly different between the quartile group by post hoc Bonferroni's test (P<0.05).

[a] Participants (196 men and 196 women) were divided into quartiles by usual diet-related GHGE separately by sex, and then combined for analysis. Usual nutrient intake and diet-related GHGE were calculated using the Multiple Source Method [42, 43]. Diet-related GHGE was adjusted for energy intake by residual method. Values are means ± SDs unless otherwise indicated.

[b] Usual diet-related GHGE (g CO$_2$-eq/d): median (25th, 75th percentiles).

[c] Trend of association was examined using a linear regression model with the median value in each quartile as a continuous variable.

[d] Probability approach was used to assess inadequacy for iron intake.

on free sugar. Of 14 nutrients with EAR, the inadequacy prevalence for protein, niacin, vitamin B-12, folate, and copper was less than 10%, while the prevalence for vitamins A, thiamine, and calcium was nearly 60%.

There was no association between diet-related GHGE and EI (Table 2). For nutrients, the diet-related GHGE was inversely associated with carbohydrate intake and positively with intakes of all the remaining nutrients, except for total fat and saturated fat. Analysis by one-

way ANOVA with *post hoc* Bonferroni test showed similar results. There was no significant difference between quartile groups for total fat, saturated fat, and free sugars. On the other hand, higher quartile groups generally had lower intake for carbohydrate but higher intakes for other nutrients than lower quartile groups.

The overall adherence to the DG and EAR was better among participants in the higher diet-related GHGE quartile compared to participants in the lower quartile (Table 3). The prevalence of inadequacy for protein, dietary fiber, potassium, vitamins A, B-6, and C, thiamine, riboflavin, calcium, magnesium, iron, and zinc decreased with increasing quartile of the diet-related GHGE. Conversely, the prevalence of inadequate sodium intake was increased with increasing quartile, but the prevalence in the lowest quartile was not low: 89% and 98% of the participants had intake above the recommendations in lowest and highest quartile group.

The diet-related GHGE was positively associated with the intakes of potato, vegetables, mushrooms, seaweeds, fish and seafood, meat, tea and coffee, and seasonings, while negatively associated with cereals, fat and oils, and sweetened beverages (Table 4). Analysis by one-way ANOVA with *post hoc* Bonferroni test also found similar results.

## Discussion

To our knowledge, this is the first study to evaluate the association between diet-related GHGE and nutritional adequacy among Japanese adults. Our result would be useful to develop future public policies or dietary guidelines to encourage sustainable healthy dietary choices. We observed overall lower prevalence of inadequacy among higher diet-related GHGE quartiles compared to lower quartiles. The participants in higher quartiles had a higher intake of almost all nutrients examined. Thus, according to the current Japanese diet, diet-related GHGE was positively associated with nutrition adequacy. Our result is inconsistent with observational studies from Western countries, where relatively consistent inverse associations between diet-related GHGE and nutritional adequacy or diet-quality scores were found [3–9] with a few exceptions [2, 7]. These inconsistent findings between this study and Western studies could be at least partly explained by difference in food intake and major food sources of diet-related GHGE. In Western countries, a few food groups such as meat and dairy products predominantly contributed to diet-related GHGE. For example, contributions of meat and dairy products were estimated at 31.3%-38.4% and 11%-25%, respectively, in European countries [14, 16, 17] and 56.6% and 18.3%, respectively, in the US [18]. Studies from the Netherlands [17], the US [6], and Ireland [14] have shown consistent positive associations between diet-related GHGE and intakes of energy-dense foods such as alcoholic beverages [14, 17], and fats [6, 14, 17] as well as meat [6, 14, 17] and dairy products [6, 14, 17]. In Japan, meat was also the top contributor to diet-related GHGE (19.6%), followed by fish/seafood (13.8%) and cereals (13.1%) [19]. Nevertheless, it should be noted that the percentage contribution of meat was lower than that in the Western countries. In addition, the contribution of dairy products (4.6%) was low [19]. In relation to diet-related GHGE, intakes of vegetables, fish/seafood, and meat showed positive associations, while those of cereals and fat and oils showed inverse associations. In this regard, intakes of nutrient-dense foods and moderate meat intake would be associated with higher intakes of protein and micronutrients as well as diet-related GHGE among Japanese, while dietary patterns in Western countries characterized by higher intake of meat and dairy products would be associated with both higher diet-related GHGE and lower nutrient intake or diet quality [3–9].

On the contrary to observational studies, previous scenario studies in European populations showed that healthy dietary pattern complying dietary guidelines would not always reduce diet-related GHGE [10–12]. These results suggest that there would be trade-offs between

**Table 3. Odds ratios for inadequate nutrient intake compared to DRIs reference value according to the quartile (Q) of usual diet-related GHGE (g CO$_2$-eq/day) among 392 Japanese adults (aged 20–69 y)[a].**

| | Diet-related GHGE | Inadequate/adequate intake participants (n)[b] | OR (95% CI) |
|---|---|---|---|
| Nutrients with Tentative Dietary Goal for Preventing Lifestyle-related Diseases | | | |
| Protein | Q1 | 54/44 | 1 (ref) |
| | Q2 | 31/67 | 0.24 (0.13-0.47) |
| | Q3 | 28/70 | 0.21 (0.11-0.41) |
| | Q4 | 12/86 | 0.07 (0.03-0.18) |
| | | | P for trend[c] <.0001 |
| Total fat | Q1 | 30/68 | 1 (ref) |
| | Q2 | 38/60 | 1.44 (0.80-2.59) |
| | Q3 | 23/75 | 0.70 (0.37-1.31) |
| | Q4 | 28/70 | 0.91 (0.49-1.68) |
| | | | P for trend[c]=0.35 |
| Saturated fat | Q1 | 71/27 | 1 (ref) |
| | Q2 | 75/23 | 1.24 (0.65-2.36) |
| | Q3 | 73/25 | 1.11 (0.59-2.09) |
| | Q4 | 76/22 | 1.31 (0.69-2.52) |
| | | | P for trend[c]=0.48 |
| Carbohydrate | Q1 | 21/77 | 1 (ref) |
| | Q2 | 24/74 | 1.19 (0.61-2.32) |
| | Q3 | 23/75 | 1.12 (0.57-2.20) |
| | Q4 | 25/73 | 1.26 (0.65-2.44) |
| | | | P for trend[c]=0.55 |
| Dietary fiber | Q1 | 94/4 | 1 (ref) |
| | Q2 | 90/8 | 0.48 (0.14-1.65) |
| | Q3 | 93/5 | 0.79 (0.21-3.04) |
| | Q4 | 83/15 | 0.24 (0.08-0.74) |
| | | | P for trend[c]=0.01 |
| Sodium | Q1 | 87/11 | 1 (ref) |
| | Q2 | 89/9 | 1.25 (0.49-3.17) |
| | Q3 | 91/7 | 1.64 (0.61-4.43) |
| | Q4 | 96/2 | 6.07 (1.31-28.15) |
| | | | P for trend[c]=0.01 |
| Potassium | Q1 | 78/20 | 1 (ref) |
| | Q2 | 65/33 | 0.51 (0.27-0.96) |
| | Q3 | 65/33 | 0.51 (0.27-0.96) |
| | Q4 | 37/61 | 0.16 (0.08-0.30) |
| | | | P for trend[c] <.0001 |
| Nutrient with the World Health Organization's conditional recommendation | | | |
| Free sugar | Q1 | 69/29 | 1 (ref) |
| | Q2 | 70/28 | 1.05 (0.57-1.95) |
| | Q3 | 73/25 | 1.23 (0.66-2.30) |
| | Q4 | 69/29 | 1.00 (0.54-1.85) |
| | | | P for trend[c]=0.92 |
| Nutrients with Estimated Average Requirement | | | |
| Vitamin A | Q1 | 75/23 | 1 (ref) |
| | Q2 | 56/42 | 0.41 (0.22-0.76) |
| | Q3 | 61/37 | 0.51 (0.27-0.94) |
| | Q4 | 49/49 | 0.31 (0.17-0.57) |
| | | | P for trend[c]=0.0006 |

(*Continued*)

**Table 3.** (Continued)

| | Diet-related GHGE | Inadequate/adequate intake participants (n)[b] | OR (95% CI) |
|---|---|---|---|
| Thiamine | Q1 | 71/27 | 1 (ref) |
| | Q2 | 61/37 | 0.63 (0.34-1.15) |
| | Q3 | 60/38 | 0.60 (0.33-1.10) |
| | Q4 | 44/54 | 0.31 (0.17-0.56) |
| | | | P for trend[c]=0.0001 |
| Riboflavin | Q1 | 48/50 | 1 (ref) |
| | Q2 | 29/69 | 0.44 (0.24-0.79) |
| | Q3 | 24/74 | 0.34 (0.18-0.62) |
| | Q4 | 15/83 | 0.19 (0.10-0.37) |
| | | | P for trend[c] <.0001 |
| Vitamin B-6 | Q1 | 47/51 | 1 (ref) |
| | Q2 | 24/74 | 0.35 (0.19-0.65) |
| | Q3 | 17/81 | 0.23 (0.12-0.44) |
| | Q4 | 8/90 | 0.10 (0.04-0.22) |
| | | | P for trend[c]<.0001 |
| Vitamin C | Q1 | 50/48 | 1 (ref) |
| | Q2 | 32/66 | 0.47 (0.26-0.83) |
| | Q3 | 28/70 | 0.38 (0.21-0.69) |
| | Q4 | 12/86 | 0.13 (0.07-0.28) |
| | | | P for trend[c] <.0001 |
| Calcium | Q1 | 72/26 | 1 (ref) |
| | Q2 | 69/29 | 0.86 (0.46-1.60) |
| | Q3 | 69/29 | 0.86 (0.46-1.60) |
| | Q4 | 58/40 | 0.52 (0.29-0.96) |
| | | | P for trend[c] =0.03 |
| Magnesium | Q1 | 58/40 | 1 (ref) |
| | Q2 | 40/58 | 0.48 (0.27-0.84) |
| | Q3 | 40/58 | 0.48 (0.27-0.84) |
| | Q4 | 23/75 | 0.21 (0.11-0.39) |
| | | | P for trend[c] <.0001 |
| Iron | Q1 | 46/52 | 1 (ref) |
| | Q2 | 33/65 | 0.51 (0.29-0.91) |
| | Q3 | 32/66 | 0.51 (0.29-0.91) |
| | Q4 | 21/77 | 0.28 (0.15-0.52) |
| | | | P for trend[c] <.0001 |
| Zinc | Q1 | 48/50 | 1 (ref) |
| | Q2 | 38/60 | 0.66 (0.37-1.16) |
| | Q3 | 36/62 | 0.61 (0.34-1.07) |
| | Q4 | 24/74 | 0.34 (0.18-0.62) |
| | | | P for trend[c]=0.0005 |

GHGE, greenhouse gas emission; $CO_2$-eq, carbon dioxide equivalents. DRIs, Dietary Reference Intakes for Japanese.

[a] Participants (196 men and 196 women) were divided into quartiles by usual diet-related GHGE separately by sex, and then combined for analysis. Usual diet-related GHGE was calculated using the Multiple Source Method (MSM) [42, 43] and then adjusted for energy intake by residual method.

[b] Inadequate intake was defined by comparing usual intake with reference values derived from Dietary Reference Intakes for Japanese 2020 except for iron intake for women aged under 50 years old and free sugar. For iron intake among women aged <50 years, less than 9.3 mg/d [50] was considered inadequate. For free sugar, the World Health Organization's conditional recommendation (<5% energy) was used.

[c] Logistic regression models were used with the median value in each quartile category of diet-related GHGE as a continuous variable.

**Table 4. Usual food intake (g/d) according to quartile (Q) of usual diet-related GHGE (g CO$_2$-eq/day) among 392 Japanese adults (aged 20–69 y)[a].**

| | All (n = 392) | Q1 (n = 98)<br>3241 (2992, 3550)[b] | Q2 (n = 98)<br>3757 (3450, 4044)[b] | Q3 (n = 98)<br>4191 (3844, 4511)[b] | Q4 (n = 98)<br>4833 (4420, 5198)[b] | P for trend[c] |
|---|---|---|---|---|---|---|
| Cereals | 449±133 | 481±155* | 459±136*† | 425±123† | 430±106† | 0.008 |
| Potatoes | 45±17 | 42±16* | 45±15* | 42±14* | 52±19† | <.0001 |
| Sugar | 15±10 | 15±11 | 14±8 | 16±10 | 16±9 | 0.43 |
| Pulses | 56±34 | 50±33* | 57±32*† | 56±33*† | 62±37† | 0.08 |
| Nuts | 3±5 | 3±5 | 3±4 | 4±5 | 3±3 | 0.88 |
| Vegetables | 245±93 | 196±72* | 238±79† | 252±79† | 296±110‡ | <.0001 |
| Fruits | 86±77 | 74±79 | 85±70 | 84±68 | 99±88 | 0.14 |
| Mushroom | 15±10 | 12±8* | 14±9*† | 16±11†‡ | 18±12‡ | <.0001 |
| Seaweeds | 5±4 | 4±3* | 5±4*† | 6±4† | 6±5† | 0.01 |
| Fish and seafood | 40±21 | 31±16* | 41±20† | 43±22† | 47±22† | <.0001 |
| Meat | 94±39 | 86±33* | 90±39*† | 97±40*† | 103±42† | 0.01 |
| Beef | 16±11 | 11±7* | 13±10*† | 17±10†† | 23±14‡ | <.0001 |
| Pork | 34±17 | 34±16 | 35±17 | 34±17 | 36±17 | 0.83 |
| Chicken | 31±15 | 31±13 | 29±15 | 33±17 | 30±15 | 0.21 |
| Processed meat products | 12±9 | 12±7 | 13±9 | 13±10 | 11±7 | 0.21 |
| Egg | 40±14 | 40±14 | 39±16 | 39±14 | 41±12 | 0.77 |
| Milk and dairy food products | 98±82 | 93±83*† | 117±93* | 86±64† | 97±82*† | 0.05 |
| Fat and oils | 21±7 | 22±7* | 22±7* | 20±7*† | 19±5† | 0.0004 |
| Confectioneries | 42±27 | 43±31 | 41±23 | 45±29 | 40±22 | 0.57 |
| Alcoholic beverages | 132±224 | 123±217 | 89±143 | 166±231 | 149±279 | 0.09 |
| Tea and coffee | 599±355 | 510±308* | 560±341* | 612±362*† | 713±380† | 0.0005 |
| Sweetened beverages | 39±73 | 58±98* | 34±59*† | 39±79*† | 24±41† | 0.01 |
| Seasonings | 119±68 | 84±34* | 106±49*† | 121±57† | 167±89‡ | <.0001 |
| Water | 515±325 | 456±266 | 511±286 | 563±374 | 531±357 | 0.13 |

GHGE, greenhouse gas emissions; CO$_2$-eq, carbon dioxide equivalents.

*†‡Maen values within a row with different symbols were significantly different between the quartile group by post hoc Bonferroni's test (P<0.05).

[a] Participants (196 men and 196 women) were divided into quartiles by usual diet-related GHGE separately by sex, and then combined for analysis. Usual food intake and diet-related GHGE were calculated using the Multiple Source Method [42, 43]. Diet-related GHGE was adjusted for energy intake by residual method. Values are means ± SDs.

[b] Usual diet-related GHGE (g CO$_2$-eq/d): median (25th-75th percentiles).

[c] Trend of association was examined using a linear regression model with the median value in each quartile as a continuous variable.

nutrient intake and diet-related GHGE when achieving further improvement of nutrition intake, although healthier diet would associate with lower diet-related GHGE within the current observed diet in Western populations. On the other hand, our study suggests that just shifting to the diets with lower diet-related GHGE that were currently observed could not achieve sufficient nutritional adequacy among Japanese. A drastic dietary change would be needed to achieve both a reduction in the diet-related GHGE and an improvement in the nutritional adequacy. Even though meat and fish were major contributors to diet-related GHGE among Japanese [19], a previous modeling study for Japanese aiming at achieving nutritional goals reported a need for increasing intake of meat and fish in young adults [53]. Thus, there would be a trade-off between impact to diet-related GHGE and contribution to micronutrient intake by animal-based foods among Japanese. It is not known what amount of intake of animal-based food should be recommended for Japanese when both nutritional benefits and impact for diet-related GHGE are taken into account. Further research is needed to

design optimized dietary patterns to achieve nutrition goals as well as reduction of diet-related GHGE.

Apart from intakes of animal-based foods, dietary modification would be needed for reducing sodium intake among Japanese. In the present study, prevalence of inadequate sodium intake was generally high and increased with increasing quartiles of diet-related GHGE. Similarly, previous Japanese studies reported that one of three or four dietary patterns identified had the lowest prevalence of inadequacy for fifteen nutrients and the highest prevalence for sodium [45, 46]. These results suggest that participants who seem to have favorable dietary patterns do not always comply with a lower sodium diet. Thus, because seasonings is the top food source of sodium (61.7% for men and 62.9% for women) [33] and the fifth largest contributor of diet-related GHGE (9.4%) among Japanese [19], reducing seasonings intake is demanded in population level for both aspects of healthy diets and diet-related GHGE.

Several limitations of the present study warrant mention. First, the participants in this study were not randomly sampled from the general Japanese population. The participants would be more health-conscious than the general Japanese population. In addition, most of the participants were working at welfare facilities. Moreover, proportion of urban and rural area was not considered at recruitment although regional differences in dietary habits were considered at sampling. Thus, further research in a national representative sample would be needed. Second, our relatively small sample size (n = 396) would limit the power to detect moderate associations with statistical significance. In addition, the estimated usual intake distributions of nutrients and foods would be uncertain due to the small sample size [54]. Nevertheless, significant associations were generally observed between diet-related GHGE and intakes of nutrient and food. Although our sample size might be sufficient to detect the difference between quartiles of diet-related GHGE, further studies with larger sample sizes would be needed. Third, the system boundaries considered in the GHGE database were limited to the production stage due to the lack of quality data to develop the GHGE database for Japanese diet including the whole life cycle. Fourth, both nutritional adequacy and diet-related GHGE were estimated based on the self-reported dietary assessment method, which is prone to measurement errors due to, for example, changes in dietary habits during the assessment period, under-recording, under-eating, and intentional, or unintentional misreporting [39]. Given that a positive association between EI and diet-related GHGE is widely reported [8, 13], misreporting of EI may have a substantial influence on the estimation of diet-related GHGE. However, diet-related GHGE was adjusted for EI by the residual method to avoid a confounding effect of energy intake on associations between diet-related GHGE and nutritional adequacy. In addition, the effect of energy-misreporting would be small in this study because similar results have shown when under- and over-reporters were excluded from the analysis (S7–S10 Tables in S1 File). Thus, any potential effect of energy misreporting was minimized in our study. Fifth, due to a lack of objective information on physical activity, bias can be present in the identification of under- and over-reporters. Nevertheless, the potential effect of energy-misreporting in this study would be small. Finally, this study was conducted from February to March. Several previous studies have reported seasonal differences in intakes among Japanese adults [55–58]. Thus, this limited period for the survey might have produced some bias in assessing the usual intake.

In conclusion, there was an overall better nutritional adequacy observed in the participants with higher diet-related GHGE when compared to individuals with lower GHGE among Japanese adults. This suggests that current dietary choices should be drastically changed to achieve sustainable diets. Further studies are needed to design sustainable diets.

## Supporting information

**S1 File.**
(DOCX)

## Acknowledgments

We thank the dietitians who supported the survey in each welfare facility for their valuable contribution and Editage (www.editage.jp) for English language editing.

## Author Contributions

**Conceptualization:** Minami Sugimoto.

**Data curation:** Keiko Asakura, Satoshi Sasaki.

**Formal analysis:** Minami Sugimoto, Aya Fujiwara.

**Funding acquisition:** Minami Sugimoto, Satoshi Sasaki.

**Investigation:** Minami Sugimoto.

**Methodology:** Minami Sugimoto.

**Project administration:** Minami Sugimoto.

**Resources:** Shizuko Masayasu.

**Supervision:** Satoshi Sasaki.

**Writing – original draft:** Minami Sugimoto.

**Writing – review & editing:** Minami Sugimoto, Kentaro Murakami, Satoshi Sasaki.

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
