## [Decision Letter · Decision Letter 0]

29 Jul 2020

PONE-D-20-15820

Association between diet-related greenhouse gas emissions and nutrient intake adequacy among Japanese adults

PLOS ONE

Dear Dr. Sasaki,

Thank you for submitting your manuscript to PLOS ONE. After careful consideration, we feel that it has merit but does not fully meet PLOS ONE’s publication criteria as it currently stands. Therefore, we invite you to submit a revised version of the manuscript that addresses the points raised during the review process.

The study is interesting and presents novel data for an understudied geographical region.  However, the authors need to make some changes before the manuscript can be published. Both reviewers indicate specific useful comments/recommendations that need to be addressed, especially concerning details in the methods section and data analysis. The writing is for the most part clear, but the ms needs to be thoroughly checked for accuracy, typos, and other mistakes.

We look forward to receiving your revised manuscript.

Kind regards,

Nicoletta Righini, PhD

Academic Editor

PLOS ONE

Journal Requirements:

2.Thank you for including the following ethics statement on the submission details page:

'This study was conducted according to the guidelines laid down in the Declaration of

Helsinki and all procedures involving research study participants were approved by the

Ethics Committee of the University of Tokyo, Faculty of Medicine (approval number:

10005, approval date: January 7, 2013). Written informed consent was obtained from

all participants.'

Please also include this information in the ethics statement in the Methods section of your manuscript.

Additional Editor Comments (if provided):

Line 165-66 - Change to: 'Because energy intake WAS highly correlated...'. 

Reviewers' comments:

Reviewer's Responses to Questions

**Comments to the Author**

1. Is the manuscript technically sound, and do the data support the conclusions?

Reviewer #1: Yes

Reviewer #2: Yes

2. Has the statistical analysis been performed appropriately and rigorously? 

Reviewer #1: No

Reviewer #2: Yes

3. Have the authors made all data underlying the findings in their manuscript fully available?

Reviewer #1: Yes

Reviewer #2: Yes

4. Is the manuscript presented in an intelligible fashion and written in standard English?

Reviewer #1: No

Reviewer #2: Yes

5. Review Comments to the Author

**Reviewer #1:** The authors described an interesting cross-sectional study aimed at investigating potential associations between diet-related GHG emissions and the diet in a convenience sample of Japanese adults. As stated by the authors, the study addresses the environmental dietary dimension which has been poorly investigated so far in the Japanese population compared to the health-related implications.

General comments

The writing is not fully clear, thus the authors should consider further editing of the English language to improve the form. Furthermore, the statistical approach that has been used lacks of post-hoc analysis that would be able to test the differences between the quartiles in which the respondents have been divided.

Specific comments

Introduction

Page 2, line 46: please make sure that the provided percentage range (i.e. 23-37%) is consistent with that indicated in the reference. It should be 21-37%.

Methods

Please rename the paragraph title as “Materials and methods”, consistently with the submission guidelines reported on the Journal website.

In this section there are some missing information, as detailed below.

Please add to the heading Page 3, line 82: the authors should specify the criteria used to determine the sample size for the recruitment (i.e. 400 people).

Page 4, lines 92-101: within the dietary assessment paragraph, please specify if the participants were trained to provide accurate dietary records. Furthermore, the authors should specify how they managed data misreporting (e.g. if the respondents were contacted after data collection and asked to provide explanation of eventual mistakes made during dietary recording).

Page 4, line 100: The authors should maybe comment on potential inconsistencies due to the use of two different food composition databases on which the energy and nutritional analyses were made.

Page 4, line 103: please amend the typo “brinks” in “drinks” and add “of” between emission and food.

Page 5, line 117: please substitute “;” with “:”.

Page 6, line 141: please delete “women” as it is repeated twice (line 140 and 141).

Page 6, lines 149-151: the authors might evaluate to move this part at the end of the “Study design and participants” section.

Page 6, lines 152-159: The authors might consider to move this part to the paragraph “Dietary assessment” for a more logical manuscript organisation.

Page 6-7: The authors should specify in the “Statistical analysis” section the statistical texts applied to compare the quartiles reported in Table 1 (e.g. 2 test).

Page 6, line 167: Please provide a brief explanation about the “residual method” used to adjust usual diet-related GHG emissions for energy.

As mentioned above in the general comments, the authors should apply a post-hoc analysis to test the differences between the quartiles in which the respondents have been divided. Indeed, the p for trend evaluation that has been used is not enough to provide such information.

Results

Page 11, line 219. Please, amend the text by substituting “carbohydrates” with “energy”. Indeed, according to the Table 2, a positive association can be observed between diet-related GHGe and CHO intake, while no association (p=0.58) can be observed for energy.

Page 11, lines 226-227: Please substitute “;” with “:” and add “the” before “lowest”.

Page 13, line 243: For consistency, please move the detail referred to the approach used to define iron intake inadequacy to line 240, as specified for free sugar intake.

Page 13, Table 4: please amend the number referred to the total sample. It should be n= 392.

Discussion

Page 14, lines 267 and 269: The sentence starting with “In contrast” is not actually opposite to the previous sentence. Please make sure about the provided statements from line 265 to line 271.

Page 15, lines 283 and 293: please add a reference.

Page 15, lines 293-295 and page 16, lines 314-317: As the sample of respondents is not representative of the whole Japanese adult population, the authors should comment on the generalisation of their findings to this population, taking into account also the background characteristics of the sample of respondents (e.g. family income, rural/urban residence, ecc).

Page 16, line 325-326: please clarify the sentence as it is not fully understandable.

**Reviewer #2:** This is an interesting, carefully planned, and well written baseline study on the relation between individual diets and associated green house gas emissions in a section of the Japanese population. It was surprising, though, that no previous baseline study was available for Japan, as noted by the authors. It was also interesting how, in the case of the population sampled, a better nutrition implied larger emissions.

I would recommend providing more information on the larger study in the Methods section, rather than just indicating that the details are published elsewhere (although both invoked references appear to be open access, at least at the time of writing this review), which will give an overview to readers, before committing to reading an additional two papers.

I would also recommend giving the manuscript a final parse to catch very few existing idiosyncratic language errors, which can be distracting (for example, insert "the" in Line 62: "Recently, THE Japanese..."; Line 72: it should be "found" rather than "founded"; Line 89: what do authors mean by "educational admission").

Under the dietary assessment more details on the way food was logged are needed: did people have to measure their food? take photographs, just write down what was consumed? Also, which equipment was provided.

Line 184: This should refer to Table 1, not table 2. In the same paragraph, please provide better definitions for under- and over-reporters.

Line 222: Should "quantile" be "quartile"?

Finally, although strictly outside of the scope of the study, but mentioned by the authors in the introduction, it would be great if they could provide some discussion about the tensions between individual choice and population-wide actions "encouraged" by public policies.

6. PLOS authors have the option to publish the peer review history of their article (what does this mean?). If published, this will include your full peer review and any attached files.

Reviewer #1: No

Reviewer #2: **Yes: **Erick de la Barrera

---

## [Author Response · Author response to Decision Letter 0]

15 Aug 2020

Reply to the Editors’ and Reviewers’ comments for the manuscript: PONE-D-20-15820

(Title: Association between diet-related greenhouse gas emissions and nutrient intake adequacy among Japanese adults)

We thank the editor and the reviewers for their very helpful comments on our paper. We have revised the manuscript by addressing each comment point-by-point as described below. All amendments and changes are highlighted in the manuscript using red font. We trust these changes will resolve any confusion and remedy the shortcomings of the paper.

Journal Requirements:

Authors’ response:

We have revised the title page to meet the style requirements. 

Page 1, line 4-23:

“Minami Sugimoto1, Kentaro Murakami2, Aya Fujiwara2, 3, Keiko Asakura4, Shizuko Masayasu5, and Satoshi Sasaki1, 2*

1 Department of Social and Preventive Epidemiology, Division of Health Sciences and Nursing, Graduate School of Medicine, University of Tokyo, Tokyo, Japan. 

2 Department of Social and Preventive Epidemiology, School of Public Health, University of Tokyo, Tokyo, Japan. 

3 Department of Nutritional Epidemiology and Shokuiku, National Institute of Biomedical Innovation, Health and Nutrition, Tokyo, Japan 

4 Department of Environmental and Occupational Health, School of Medicine, Toho University, Tokyo, Japan. 

5Ikurien-Naka, Ibaraki 311-0105, Japan.

*Corresponding author

E-mail: stssasak@m.u-tokyo.ac.jp; sasakicrs@m.u-tokyo.ac.jp

Page 4, line 86:

Materials and Methods

2.Thank you for including the following ethics statement on the submission details page:

'This study was conducted according to the guidelines laid down in the Declaration of Helsinki and all procedures involving research study participants were approved by the Ethics Committee of the University of Tokyo, Faculty of Medicine (approval number: 10005, approval date: January 7, 2013). Written informed consent was obtained from all participants.'

Please also include this information in the ethics statement in the Methods section of your manuscript.

Authors’ response:

We have added the following information in the Method section.

Page 4, line 107-110:

This study was conducted according to the guidelines laid down in the Declaration of Helsinki and all procedures involving research study participants were approved by the Ethics Committee of the University of Tokyo, Faculty of Medicine (approval number: 10005, approval date: January 7, 2013). Written informed consent was obtained from all participants. 

Authors’ response:

We have added questions to assess demographic variables including occupation, educational background, and smoking habits as “S1 Appendix.” (see S1 Appendix in “Supporting information”)　We have not included a questionnaire because the primary focus of our manuscript is not to develop a questionnaire or assessment tool. 

Authors’ response:

We have shown the analytical results by sex and results excluded under- and over- reporters in the supporting material. (see Revised Supplemental Tables, S2-S10).

Page 7, line 189-190:

…and the analysis by sex showed similar results (S2-S6 Tables).

Page 18, line 368-370:

In addition, the effect of energy-misreporting would be small in this study because similar results have shown when under- and over-reporters were excluded from the analysis (S7-S10 Tables).

Page 25, line 559-581:

Supporting information

S1 Appendix. Questions for demographic variables

S1 Table. Sex- and age-specific Diet reference intakes values for Japanese population

S2 Table. Basic characteristics of participants according to quartile (Q) of diet-related GHGE (g CO2-eq/day) among 196 Japanese men and 196 women (aged 20-69 y)

S3 Table. Usual nutrient intake according to quartile (Q) of diet-related GHGE (g CO2-eq/day) among 196 Japanese men (aged 20-69 y)

S4 Table. Usual nutrient intake according to quartile (Q) of diet-related GHGE (g CO2-eq/day) among 196 Japanese women (aged 20-69 y)

S5 Table. Odds ratios for inadequate nutrient intake compared to DRI reference value according to the quartile (Q) of usual diet-related GHGE (g CO2-eq/day) among 196 Japanese men and 196 women (aged 20-69 y)

S6 Table. Usual food intake (g/d) according to quartile (Q) of usual diet-related GHGE (g CO2-eq/day) among 196 Japanese men and 196 women (aged 20-69 y)

S7 Table. Basic characteristics of participants according to quartile (Q) of diet-related GHGE (g CO2-eq/day) among 369 Japanese adults with plausible energy intake

S8 Table. Usual nutrient intake according to quartile (Q) of diet-related GHGE (g CO2-eq/day) among 369 Japanese adults (aged 20-69 y) with plausible energy intake

S9 Table. Odds ratios for inadequate nutrient intake compared to DRI reference value according to the quartile (Q) of usual diet-related GHGE (g CO2-eq/day) among 369 Japanese adults (aged 20-69 y) with plausible energy intake

S10 Table. Usual food intake (g/d) according to quartile (Q) of usual diet-related GHGE (g CO2-eq/day) among 369 Japanese adults (aged 20-69 y) with plausible energy intake

Additional Editor Comments (if provided):

Line 165-66 - Change to: 'Because energy intake WAS highly correlated...'. 

Authors’ response:

We have revised the text.

Page 7, line 183-184:

Because energy intake was highly correlated…

Reviewer #1: 

The authors described an interesting cross-sectional study aimed at investigating potential associations between diet-related GHG emissions and the diet in a convenience sample of Japanese adults. As stated by the authors, the study addresses the environmental dietary dimension which has been poorly investigated so far in the Japanese population compared to the health-related implications.

Authors’ response:

Thank you for your comments.

General comments

The writing is not fully clear, thus the authors should consider further editing of the English language to improve the form. Furthermore, the statistical approach that has been used lacks of post-hoc analysis that would be able to test the differences between the quartiles in which the respondents have been divided.

Authors’ response:

We have revised the English language as follows. In addition, we have added post-hoc analysis in Tables 1, 2 and 4 and accordingly made a number of revisions in “Materials and methods” and “Results” sections as shown below. (also see Tables 1, 2, and 4 in the revised manuscript)

Grammatical corrections:

Page 3, line 60-62:

However, these previous studies mainly come from Western countries [2-18], while research is limited among Asian countries including Japan, where meat intake is lower than Western countries [11–13].

Page 3, line 70:

Recently, the Japanese diet has been characterized by…

Page 3, line 79-81:

…while negative or no associations between meat intake and mortality from cardiovascular disease or cancer were generally found in Asian population with low meat intake [31,32].

Page 6, line 153-155:

The usual percent of energy from protein, fat, carbohydrate, saturated fat, and free sugar was estimated by the MSM after calculation of percent energy from these nutrients for each assessment day.

Page 7, line 183-184:

Because energy intake was highly correlated with both nutrient intakes and diet-related GHGE,…

Page 12, line 249-251:

There was no association between diet-related GHGE and EI (Table 2). For nutrients, the diet-related GHGE was inversely associated with carbohydrate intake and positively with intakes of all the remaining nutrients, except for total fat and saturated fat.

Page 16, line 331: 

Western populations

Page 17, line 349-352: 

Thus, because seasonings is the top food source of sodium (61.7% for men and 62.9% for women) [33] and the fifth largest contributor of diet-related GHGE (9.4%) among Japanese [11], reducing seasonings intake is demanded in population level for both aspects of healthy diets and diet-related GHGE.

Page 18, line 365:

misreporting of EI may have a substantial influence

Additional statistical test:

Page 7, line 193-194:

Mean differences of age and BMI among quartile groups were also tested by one-way analysis of variance (ANOVA) with the post hoc Bonferroni test.

Page 8, line 198-199:

Differences between quartiles were also assessed using one-way ANOVA with post hoc Bonferroni’s test.

Page 8, line 205-207:

In addition, the association between diet-related GHGE and the usual food intake was examined using linear regression models and one-way ANOVA with post hoc Bonferroni’s test.

Page 9, line 219-220:

*†‡Maen values within a row with different symbols were significantly different between the quartile group by post hoc Bonferroni’s test (P<0.05).

Page 11, line 240:

*†‡Maen values within a row with different symbols were significantly different between the quartile group by post hoc Bonferroni’s test (P<0.05).

Page 12, line 252-255:

There was no significant difference between quartile groups for total fat, saturated fat, and free sugars. On the other hand, higher quartile groups generally had lower intake for carbohydrate but higher intakes for other nutrients than lower quartile groups.

Page 14, line 282-283:

Analysis by one-way ANOVA with post hoc Bonferroni test also found similar results.

Page 15, line 287-288:

*†‡Maen values within a row with different symbols were significantly different between the quartile group by post hoc Bonferroni’s test (P<0.05).

Specific comments

Introduction

Page 2, line 46: please make sure that the provided percentage range (i.e. 23-37%) is consistent with that indicated in the reference. It should be 21-37%.

Authors’ response:

We have checked the reference and revised the percentage range as suggested.

Page 2, line 53: 

which contribute 21-37% global

Methods

Please rename the paragraph title as “Materials and methods”, consistently with the submission guidelines reported on the Journal website.

Authors’ response:

We have revised the paragraph title.

Page 4, line 85: 

Materials and Methods

In this section there are some missing information, as detailed below.

Please add to the heading Page 3, line 82: the authors should specify the criteria used to determine the sample size for the recruitment (i.e. 400 people).

Authors’ response:

In accordance with the comment, we have made the following revision. 

Page 4, line 87-106:

This cross-sectional study was based on data from healthy Japanese adults aged 20–69 years. Data collection was conducted in 20 study areas covering 23 of 47 prefectures between February and March 2013. Details of the study have been reported elsewhere [33,34]. The primary objective of this survey was to estimate sodium and potassium excretion using biomarker and to identify food sources of sodium and potassium. First, 199 dietitians working in separate welfare facilities were recruited as research dietitians supporting the survey. Next, the research dietitians recruited participants from their co-workers or family members of co-workers with stratifying by sex and by five 10-year age bands (20–29, 30–39, 40– 49, 50–59, and 60–69 years). The number of participants was targeted to be 40 adults in each study area to allow for statistical analysis stratified by sex, age, body mass index (BMI; in kg/m2), and physical activity. The exclusion criteria were: (i) licensed dietary or medical provider, (ii) residence in the prefecture or adjacent prefecture in which the facility was located for less than 6 months, (iii) individuals who were under diet therapy prescribed by a doctor or dietitian at the time of the study or within 1 y before the study, (iv) pregnant or lactating women, and (v) individuals who had history of hospitalization for diabetes education. Of those 800 adults recruited, nine adults withdrew from the survey. In total, 791 adults participated. To reduce the burden to the participants and the research dietitians, half of the participants (n=400) were also asked to complete diet records. A total of 392 adults (196 men and 196 women) completed diet records and were included in the present analysis. BMI was calculated based on measured weight and height. The participants’ occupation, educational background, and smoking habits were assessed using a questionnaire (S1 Appendix).

Page 4, lines 92-101: within the dietary assessment paragraph, please specify if the participants were trained to provide accurate dietary records. Furthermore, the authors should specify how they managed data misreporting (e.g. if the respondents were contacted after data collection and asked to provide explanation of eventual mistakes made during dietary recording).

Authors’ response:

In accordance with the comment, we have made the following revision. 

Page 5, line 112- page 6, 120:

Dietary intake was assessed by four-non-consecutive-day diet records. The assessment days consisted of three working days and one day off. All participants were provided with digital kitchen scale (KD-812WH; Tanita, Tokyo, Japan), measuring spoon, measuring cup, a manual for the diet record, and recording sheets and instructed how to weigh and record foods and beverages consumed. Each participant was asked to weigh and record all food and beverages consumed on the four assessment days using the provided equipments and recording sheet. When weighing was difficult (e.g. eating out), the restaurant’s name, name of dishes, and an estimated amount of leftovers were reported. Pictures of food and beverages were also provided by some participants but not mandatory.

Page 5, line 123-124:

The research dietitian contacted the participants to clarify any ambiguities or missing data in the recording sheets.

Page 4, line 100: The authors should maybe comment on potential inconsistencies due to the use of two different food composition databases on which the energy and nutritional analyses were made.

Authors’ response:

Free sugar intake was not calculated based on an independent database from the “Standard Tables of Food Composition in Japan” but based on the “Standard Tables of Food Composition in Japan” with additional values for free sugar content. Fujiwara et al [ref 37 and 38] added free sugar content to each item in the “Standard Tables of Food Composition in Japan” based on published sources because the “Standard Tables of Food Composition in Japan” does not originally include the content of free sugars. We have made the following revision to make it clear.

Page 5, line 124-127:

Daily intakes of foods, energy, and nutrients were estimated based on the Standard Tables of Food Composition in Japan in which free sugar content of each food item was added based on published sources [37,38] due to the insufficiency of data for free sugars in original food composition tables.

Page 4, line 103: please amend the typo “brinks” in “drinks” and add “of” between emission and food.

Authors’ response:

We have revised the text in the manuscript.

Page 5, line 137:

Greenhouse gas emissions of food and drinks

Page 5, line 117: please substitute “;” with “:”.

Authors’ response:

We have revised the text in the manuscript.

Page 6, line 150:

…as following three steps: calculating…

Page 6, line 141: please delete “women” as it is repeated twice (line 140 and 141).

Authors’ response:

We have revised the text in the manuscript.

Page 7, line 172-173:

…among women under 50 years old women using…

Page 6, lines 149-151: the authors might evaluate to move this part at the end of the “Study design and participants” section.

Authors’ response:

Thank you for the comments. We moved them at the end of the “Study design and participants” section as suggested.

Page 5, line 104-106:

BMI was calculated based on measured weight and height. The participants’ occupation, educational background, and smoking habits were assessed using a questionnaire (S1 Appendix).

Page 6, lines 152-159: The authors might consider to move this part to the paragraph “Dietary assessment” for a more logical manuscript organisation.

Authors’ response:

We have moved this part to “Dietary assessment.”

Page 5, line 128-136:

To evaluate the accuracy of the reported energy intake (EI), the ratio of EI to basal metabolic rate (BMR) (EI:BMR) was compared to the Goldberg cut-off [39]. For this, the average EI of the four-assessment-days was used. BMR was calculated using the sex-specific equation for Japanese population based on age, weight, and height. Assuming sedentary lifestyle for all subjects because of a lack of objective information on physical activity in this study (physical activity level: 1.55 for men and 1.56 for women), under-, plausible-, and over-reporters were defined as having EI:BMR <1.02, 1.02-2.35, and >2.35 for men, and <1.03, 1.03-2.36, and >2.36 for women, respectively. Under- and over- reporters were identified in this study but were not excluded from the analysis to avoid bias for that exclusion [40,41]. 

Page 6-7: The authors should specify in the “Statistical analysis” section the statistical texts applied to compare the quartiles reported in Table 1 (e.g. 2 test).

Authors’ response:

In accordance with the comment, we have made the following revision in the “Materials and Methods” section.

Page 7, line 191-195:

Linear regression models were constructed to examine the association between diet-related GHGE (g CO2-eq/d) and age and BMI using the median value for each quantile as a continuous variable. Mean differences of age and BMI among quartile groups were also tested by one-way analysis of variance (ANOVA) with the post hoc Bonferroni test. The chi-square test was used to test differences in living area, occupation, educational background, and smoking habits.

Page 6, line 167: Please provide a brief explanation about the “residual method” used to adjust usual diet-related GHG emissions for energy.

Authors’ response:

We have added the explanation about the “residual method.”

Page 7, line 185-187:

Energy-adjusted diet-related GHGE was computed as the residuals from the regression model, with total energy intake as the independent variable and absolute diet-related GHGE as the dependent variable [53].

As mentioned above in the general comments, the authors should apply a post-hoc analysis to test the differences between the quartiles in which the respondents have been divided. Indeed, the p for trend evaluation that has been used is not enough to provide such information.

Authors’ response:

In accordance with the comment, we have added post-hoc analysis and accordingly made revision in “Materials and methods” and “Results” sections, and Tables 1, 2, and 4. (see Tables 1, 2, and 4 in the revised manuscript)

Page 7, line 193-194:

Mean differences of age and BMI among quartile groups were also tested by one-way analysis of variance (ANOVA) with the post hoc Bonferroni test.

Page 8, line 198-199:

Differences between quartiles were also assessed using one-way ANOVA with post hoc Bonferroni’s test.

Page 8, line 205-207:

In addition, the association between diet-related GHGE and the usual food intake was examined using linear regression models and one-way ANOVA with post hoc Bonferroni’s test.

Page 9, line 219-220:

*†‡Maen values within a row with different symbols were significantly different between the quartile group by post hoc Bonferroni’s test (P<0.05).

Page 11, line 240:

*†‡Maen values within a row with different symbols were significantly different between the quartile group by post hoc Bonferroni’s test (P<0.05).

Page 12, line 252-255:

There was no significant difference between quartile groups for total fat, saturated fat, and free sugars. On the other hand, higher quartile groups generally had lower intake for carbohydrate but higher intakes for other nutrients than lower quartile groups.

Page 14, line 282-283:

Analysis by one-way ANOVA with post hoc Bonferroni test also found similar results.

Page 15, line 287-288:

*†‡Maen values within a row with different symbols were significantly different between the quartile group by post hoc Bonferroni’s test (P<0.05).

Results

Page 11, line 219. Please, amend the text by substituting “carbohydrates” with “energy”. Indeed, according to the Table 2, a positive association can be observed between diet-related GHGe and CHO intake, while no association (p=0.58) can be observed for energy.

Authors’ response:

In accordance with the comment, we have made the following revision

Page 13, line 249-251:

There was no association between diet-related GHGE and EI (Table 2). For nutrients, the diet-related GHGE was inversely associated with carbohydrate intake and positively with intakes of all the remaining nutrients, except for total fat and saturated fat.

Page 11, lines 226-227: Please substitute “;” with “:” and add “the” before “lowest”.

Authors’ response:

We have revised the manuscript.

Page 12, line 261-262:

the prevalence in the lowest quartile was not low: 89% and 98% of the participants had intake above the recommendations in lowest and highest quartile group.

Page 13, line 243: For consistency, please move the detail referred to the approach used to define iron intake inadequacy to line 240, as specified for free sugar intake.

Authors’ response:

We have added the detail referred to the approach to define inadequacy of iron intake and have revised the next sentence accordingly. To avoid repetition, we deleted the footnote “d.”

Page 14, line 271-277:

b Inadequate intake was defined by comparing usual intake with reference values derived from Dietary Reference Intakes for Japanese 2020 except for iron intake for women under 50 years old and free sugar. For iron intake among women aged <50 years, less than 9.3 mg/d [51] was considered inadequate. For free sugar, the World Health Organization’s conditional recommendation (<5% energy) was used.

c Logistic regression models were used with the median value in each quartile category of diet-related GHGE as a continuous variable.

d Probability approach was used to assess inadequacy for iron intake.

Page 13, Table 4: please amend the number referred to the total sample. It should be n= 392.

Authors’ response:

We have revised the number of the total sample in Table 4. (See Table 4 in the revised the manuscript)

Discussion

Page 14, lines 267 and 269: The sentence starting with “In contrast” is not actually opposite to the previous sentence. Please make sure about the provided statements from line 265 to line 271.

Authors’ response:

We have revised the words as follows.

Page 15, line 304-307:

Thus, according to the current Japanese diet, diet-related GHGE was positively associated with nutrition adequacy. Our result is inconsistent with observational studies from Western countries, where relatively consistent inverse associations between diet-related GHGE and nutritional adequacy or diet-quality scores were found [3,14–19] with a few exceptions [2,17].

Page 15, lines 283 and 293: please add a reference.

Authors’ response:

We have added references to the latter part of this sentence. We did not add a reference in the former part because it was a summary of our study. 

Page 16, line 321-325:

In this regard, intakes of nutrient-dense foods and moderate meat intake would be associated with higher intakes of protein and micronutrients as well as diet-related GHGE among Japanese, while dietary patterns in Western countries characterized by higher intake of meat and dairy products would be associated with both higher diet-related GHGE and lower nutrient intake or diet quality [3,14–19].

Page 15, lines 293-295 and page 16, lines 314-317: As the sample of respondents is not representative of the whole Japanese adult population, the authors should comment on the generalisation of their findings to this population, taking into account also the background characteristics of the sample of respondents (e.g. family income, rural/urban residence, ecc).

Authors’ response:

The generalizability of our sample was low because most of them were working at the welfare facilities. In addition, the proportion of urban or rural residence and educational background was not considered at the recruitment. Further, information for family income was not obtained. Thus, we have revised the manuscript as shown below.

Page 17, line 354- 358:

The participants would be more health-conscious than the general Japanese population. In addition, most of the participants were working at welfare facilities. Moreover, proportion of urban and rural area was not considered at recruitment although regional differences in dietary habits were considered at sampling. Thus, further research in a national representative sample would be needed.

Page 16, line 325-326: please clarify the sentence as it is not fully understandable.

Authors’ response:

We have revised the manuscript and added the results when under- and over-reporters were excluded as Supplemental Tables. (see the Tables in revised “Supporting information”)

Page 18, line 366-370:

However, diet-related GHGE was adjusted for EI by the residual method to avoid a confounding effect of energy intake on associations between diet-related GHGE and nutritional adequacy. In addition, the effect of energy-misreporting would be small in this study because similar results have shown when under- and over-reporters were excluded from the analysis (S7-S10 Tables).

Reviewer #2: 

This is an interesting, carefully planned, and well written baseline study on the relation between individual diets and associated green house gas emissions in a section of the Japanese population. It was surprising, though, that no previous baseline study was available for Japan, as noted by the authors. It was also interesting how, in the case of the population sampled, a better nutrition implied larger emissions.

Authors’ response:

Thank you for your comments.

I would recommend providing more information on the larger study in the Methods section, rather than just indicating that the details are published elsewhere (although both invoked references appear to be open access, at least at the time of writing this review), which will give an overview to readers, before committing to reading an additional two papers.

Authors’ response:

We have added more information for “Study design and participants” and “Dietary assessment” sections.

Page 4, line 87-104:

Study design and participants

This cross-sectional study was based on data from healthy Japanese adults aged 20–69 years. Data collection was conducted in 20 study areas covering 23 of 47 prefectures between February and March 2013. Details of the study have been reported elsewhere [33,34]. The primary objective of this survey was to estimate sodium and potassium excretion using biomarker and to identify food sources of sodium and potassium. First, 199 dietitians working in separate welfare facilities were recruited as research dietitians supporting the survey. Next, the research dietitians recruited participants from their co-workers or family members of co-workers with stratifying by sex and by five 10-year age bands (20–29, 30–39, 40– 49, 50–59, and 60–69 years). The number of participants was targeted to be 40 adults in each study area to allow for statistical analysis stratified by sex, age, body mass index (BMI; in kg/m2), and physical activity. The exclusion criteria were: (i) licensed dietary or medical provider, (ii) residence in the prefecture or adjacent prefecture in which the facility was located for less than 6 months, (iii) individuals who were under diet therapy prescribed by a doctor or dietitian at the time of the study or within 1 y before the study, (iv) pregnant or lactating women, and (v) individuals who had history of hospitalization for diabetes education. Of those 800 adults recruited, nine adults withdrew from the survey. In total, 791 adults participated. To reduce the burden to the participants and the research dietitians, half of the participants (n=400) were also asked to complete diet records. A total of 392 adults (196 men and 196 women) completed diet records and were included in the present analysis.

Page 4, line 112-page 5, 127:

Dietary intake was assessed by four-non-consecutive-day diet records. The assessment days consisted of three working days and one day off. All participants were provided with digital kitchen scale (KD-812WH; Tanita, Tokyo, Japan), measuring spoon, measuring cup, a manual for the diet record, and recording sheets and instructed how to weigh and record foods and beverages consumed. Each participant was asked to weigh and record all food and beverages consumed on the four assessment days using the provided equipments and recording sheet. When weighing was difficult (e.g. eating out), the restaurant’s name, name of dishes, and an estimated amount of leftovers were reported. Pictures of food and beverages were also provided by some participants but not mandatory. All recorded foods and beverages were assigned food item numbers according to the Standard Tables of Food Composition in Japan, Fifth Revised and Enlarged Edition [36]. All records were checked twice (by the research dietitians at each facility and trained dietitian staff at the survey center). The research dietitian contacted the participants to clarify any ambiguities or missing data in the recording sheets. Daily intakes of foods, energy, and nutrients were estimated based on the Standard Tables of Food Composition in Japan in which free sugar content of each food item was added based on published sources [37,38] due to the insufficiency of data for free sugars in original food composition tables.

I would also recommend giving the manuscript a final parse to catch very few existing idiosyncratic language errors, which can be distracting (for example, insert "the" in Line 62: "Recently, THE Japanese..."; Line 72: it should be "found" rather than "founded"; 

We have revised the language errors as follows.

Page 3, line 70:

Recently, the Japanese diet has been characterized by…

Page 3, line 79-81:

…while negative or no associations between meat intake and mortality from cardiovascular disease or cancer were generally found in Asian population with low meat intake [31,32]

Line 89: what do authors mean by "educational admission").

Authors’ response:

"Educational admission" is a kind of hospitalization for patients especially of diabetes with poor glycemic control. During hospitalization, patients are taught about proper treatment for their condition, motivate them to improve their lifestyle habits. We have made a revision in the text not to use the word "educational admission."

Page 4, 100-101:

(v) individuals who had history of hospitalization for diabetes education.

Under the dietary assessment more details on the way food was logged are needed: did people have to measure their food? take photographs, just write down what was consumed? Also, which equipment was provided.

Authors’ response:

We have added information about the dietary assessment. They measured their food and just write down them. Some participants took pictures of food and attached them as supporting information about food although it was not mandatory.

Page 4, line 112-page 5, 127:

Dietary intake was assessed by four-non-consecutive-day diet records. The assessment days consisted of three working days and one day off. All participants were provided with digital kitchen scale (KD-812WH; Tanita, Tokyo, Japan), measuring spoon, measuring cup, a manual for the diet record, and recording sheets and instructed how to weigh and record foods and beverages consumed. Each participant was asked to weigh and record all food and beverages consumed on the four assessment days using the provided equipments and recording sheet. When weighing was difficult (e.g. eating out), the restaurant’s name, name of dishes, and an estimated amount of leftovers were reported. Pictures of food and beverages were also provided by some participants but not mandatory. All recorded foods and beverages were assigned food item numbers according to the Standard Tables of Food Composition in Japan, Fifth Revised and Enlarged Edition [36]. All records were checked twice (by the research dietitians at each facility and trained dietitian staff at the survey center). The research dietitian contacted the participants to clarify any ambiguities or missing data in the recording sheets. Daily intakes of foods, energy, and nutrients were estimated based on the Standard Tables of Food Composition in Japan in which free sugar content of each food item was added based on published sources [37,38] due to the insufficiency of data for free sugars in original food composition tables.

Line 184: This should refer to Table 1, not table 2. In the same paragraph, please provide better definitions for under- and over-reporters.

Authors’ response:

We have revised the text. We also added the definition of under- and over- reporters although they were described in the “Method and material” section.

Page 8, line 209:

The basic characteristics of the participants are described in Table 1.

Page 8, line 211-213:

The prevalence of under-reporters (defined as having EI:BMR <1.02 and for men and <1.03 for women) was 3.6% (nine men and five women) and that of over-reporters (defined as having EI:BMR >2.35 for men and >2.36 for women) was 2.3% (three men and six women).

Line 222: Should "quantile" be "quartile"?

Authors’ response:

We have revised the text.

Page 12, line 256-262:

The overall adherence to the DG and EAR was better among participants in the higher diet-related GHGE quartile compared to participants in the lower quartile (Table 3). The prevalence of inadequacy for protein, dietary fiber, potassium, vitamins A, B-6, and C, thiamine, riboflavin, calcium, magnesium, iron, and zinc decreased with increasing quartile of the diet-related GHGE. Conversely, the prevalence of inadequate sodium intake was increased with increasing quartile, but the prevalence in the lowest quartile was not low: 89% and 98% of the participants had intake above the recommendations in lowest and highest quartile group.

Finally, although strictly outside of the scope of the study, but mentioned by the authors in the introduction, it would be great if they could provide some discussion about the tensions between individual choice and population-wide actions "encouraged" by public policies.

Authors’ response:

In accordance with your comment, we have shortly discussed the tensions between individual choice and recommendation in the “Discussion” section.

Page 16, line 300-301:

To our knowledge, this is the first study to evaluate the association between diet-related GHGE and nutritional adequacy among Japanese adults. Our result would be useful to develop future public policies or dietary guidelines to encourage sustainable healthy dietary choices.

---

## [Decision Letter · Decision Letter 1]

23 Sep 2020

PONE-D-20-15820R1

Association between diet-related greenhouse gas emissions and nutrient intake adequacy among Japanese adults

PLOS ONE

Dear Dr. Sasaki,

Thank you for submitting your manuscript to PLOS ONE. After careful consideration, we feel that it has merit but does not fully meet PLOS ONE’s publication criteria as it currently stands. Therefore, we invite you to submit a revised version of the manuscript that addresses the points raised during the review process.

The authors did a good job responding to the reviewers' queries and now the manuscript has significantly improved. Please just take care of a few minor comments, after which the ms can be accepted.

We look forward to receiving your revised manuscript.

Kind regards,

Nicoletta Righini, PhD

Academic Editor

PLOS ONE

Additional Editor Comments (if provided):

Please check the spelling of MEAN in several Tables (e.g., 1, 4..). Currently it appears as MAEN

Reviewers' comments:

Reviewer's Responses to Questions

**Comments to the Author**

1. If the authors have adequately addressed your comments raised in a previous round of review and you feel that this manuscript is now acceptable for publication, you may indicate that here to bypass the “Comments to the Author” section, enter your conflict of interest statement in the “Confidential to Editor” section, and submit your "Accept" recommendation.

Reviewer #1: (No Response)

2. Is the manuscript technically sound, and do the data support the conclusions?

Reviewer #1: Yes

3. Has the statistical analysis been performed appropriately and rigorously? 

Reviewer #1: Yes

4. Have the authors made all data underlying the findings in their manuscript fully available?

Reviewer #1: Yes

5. Is the manuscript presented in an intelligible fashion and written in standard English?

Reviewer #1: Yes

6. Review Comments to the Author

**Reviewer #1:** I was a reviewer also of the first version of the manuscript. The authors significantly improved the quality of the manuscript and addressed the previous comments. Only minor revisions remain to be considered before publication.

Introduction

Page 3, line 65-66: The sentence “The dietary aspects have not been included in this statement nor mentioned in the dietary guidelines” is understandable, however it could be slightly modified to increase the readability and clarity by substituting “The dietary aspects” with, for example, “the dietary environmental dimension”.

Page 3, line 70-73: the authors should consider to change a bit the sentence to improve the form by substituting “while” (line 71) with “and” or deciding to divide this long sentence in two parts.

Material and methods

Thank you for adding information about the criteria used to determine the sample size and for providing details on the larger study. Thank you also for providing information about data collection and data management. This information is needed for data replicability and study clarity. Furthermore, thank you for adding the post hoc analysis to compare the quartile groups.

Page 7, line 192 and page 8, line 200: Should “quantile” be “quartile”?

Discussion

Page 16, line 316-319: please check the accuracy of the sentence. Meat contribution to diet-related to GHGE has been indicated as 19.6%. This percentage is the highest compared to those referred to the other food groups. As a consequence, meat contribution should not be mentioned together with dairy products (4.6%), but together with cereals (13.1%), vegetables/fruits (7.6%), and fish/seafood (13.8%). Once rectified, the sentence will be compliant with what properly mentioned at page 17, line 334 and 335.

Page 18, line 368-370: The authors should mention the lack of objective information on physical activity as a limitation of the study in the discussion section. Indeed, bias can be present in the identification of under- and over- reporters, even though potential effect of energy-misreporting would be small. Another limitation is the relatively limited sample size that should be highlighted.

7. PLOS authors have the option to publish the peer review history of their article (what does this mean?). If published, this will include your full peer review and any attached files.

Reviewer #1: No

---

## [Author Response · Author response to Decision Letter 1]

29 Sep 2020

Reply to the Editors’ and Reviewers’ comments for the manuscript: PONE-D-20-15820R1

(Title: Association between diet-related greenhouse gas emissions and nutrient intake adequacy among Japanese adults)

We thank the editor and the reviewers for their very helpful comments on our paper. We have revised the manuscript by addressing each comment point-by-point as described below. All amendments and changes are highlighted in the manuscript using a red font. We trust these changes will resolve any confusion and remedy the shortcomings of the paper.

Reviewer #1: 

I was a reviewer also of the first version of the manuscript. The authors significantly improved the quality of the manuscript and addressed the previous comments. Only minor revisions remain to be considered before publication.

Authors’ response:

Thank you for your comments.

Introduction

Page 3, line 65-66: The sentence “The dietary aspects have not been included in this statement nor mentioned in the dietary guidelines” is understandable, however it could be slightly modified to increase the readability and clarity by substituting “The dietary aspects” with, for example, “the dietary environmental dimension”.

Authors’ response:

Based on your comments, we have revised the text as follows.

“The environmental dimension of diet has not been included in this statement nor mentioned in the dietary guidelines.” (page 3, line 65-66)

Page 3, line 70-73: the authors should consider to change a bit the sentence to improve the form by substituting “while” (line 71) with “and” or deciding to divide this long sentence in two parts.

Authors’ response:

In accordance with the comment, we have decided to divide this sentence into two parts; one for food consumption and another for nutrient intakes.

“The contemporary Japanese diet is typically high in refined grains, seaweeds, vegetables, fish, legumes and low in whole grains, nuts and seeds, dairy products, sugar-sweetened beverage, and processed and unprocessed red meats [27–29]. At the nutrient level, it is characterized by a high intake of sodium and a low intake of dietary fiber, calcium, and saturated fat [29,30].” (page 3, line 70-73)

Material and methods

Thank you for adding information about the criteria used to determine the sample size and for providing details on the larger study. Thank you also for providing information about data collection and data management. This information is needed for data replicability and study clarity. Furthermore, thank you for adding the post hoc analysis to compare the quartile groups.

Authors’ response:

Thank you for your comments.

Page 7, line 192 and page 8, line 200: Should “quantile” be “quartile”?

Authors’ response:

Thank you for finding our careless typographical errors, which have been corrected as shown below. 

“using the median value for each quartile” (page 7, line 195)

“using the median value for each quartile” (page 8, line 200)

“inadequacy were calculated for each quartile of diet-related GHGE” (page 8, line 203)

“c Trend of association was examined using a linear regression model with the median value in each quartile as a continuous variable” (page 16, line 297-298)

Discussion

Page 16, line 316-319: please check the accuracy of the sentence. Meat contribution to diet-related to GHGE has been indicated as 19.6%. This percentage is the highest compared to those referred to the other food groups. As a consequence, meat contribution should not be mentioned together with dairy products (4.6%), but together with cereals (13.1%), vegetables/fruits (7.6%), and fish/seafood (13.8%). Once rectified, the sentence will be compliant with what properly mentioned at page 17, line 334 and 335.

Authors’ response:

In accordance with the comment, we have made the following revision. In line with the revision of the sentences that you pointed out, we have revised some sentences in the same paragraph.

“In Japan, meat was also the top contributor to diet-related GHGE (19.6%), followed by fish/seafood (13.8%) and cereals (13.1%)[11]. Nevertheless, it should be noted that the percentage contribution of meat was lower than that in the Western countries. In addition, the contribution of dairy products (4.6%) was low [11]. In relation to diet-related GHGE, intakes of vegetables, fish/seafood, and meat showed positive associations, while those of cereals and fat and oils showed inverse associations.” (page 17, line 319-323)

Page 18, line 368-370: The authors should mention the lack of objective information on physical activity as a limitation of the study in the discussion section. Indeed, bias can be present in the identification of under- and over- reporters, even though potential effect of energy-misreporting would be small. Another limitation is the relatively limited sample size that should be highlighted.

Authors’ response:

In accordance with the comment, we have added the limitation for sample size and a lack of observational measure for physical activity level.

“Second, our relatively small sample size (n = 396) would limit the power to detect moderate associations with statistical significance. In addition, the estimated usual intake distributions of nutrients and foods would be uncertain due to the small sample size [54]. Nevertheless, significant associations were generally observed between diet-related GHGE and intakes of nutrient and food. Although our sample size might be sufficient to detect the difference between quartiles of diet-related GHGE, further studies with larger sample sizes would be needed. Third,…” (page 17, line 360- page 18, 366)

“Fourth, both nutritional adequacy and diet-related…” (page 18, 368)

“Fifth, due to a lack of objective information on physical activity, bias can be present in the identification of under- and over-reporters. Nevertheless, the potential effect of energy-misreporting in this study would be small.” (page 18, line 378-380)

---

## [Editor Report · Decision Letter 2]

5 Oct 2020

Association between diet-related greenhouse gas emissions and nutrient intake adequacy among Japanese adults

PONE-D-20-15820R2

Dear Dr. Sasaki,

We’re pleased to inform you that your manuscript has been judged scientifically suitable for publication and will be formally accepted for publication once it meets all outstanding technical requirements.

Kind regards,

Nicoletta Righini, PhD

Academic Editor

PLOS ONE

---

## [Editor Report · Acceptance letter]

14 Oct 2020

PONE-D-20-15820R2 

Association between diet-related greenhouse gas emissions and nutrient intake adequacy among Japanese adults 

Dear Dr. Sasaki:

I'm pleased to inform you that your manuscript has been deemed suitable for publication in PLOS ONE. Congratulations! Your manuscript is now with our production department. 

Kind regards, 

on behalf of

Dr. Nicoletta Righini 

Academic Editor

PLOS ONE